# Training on more Reachable Tasks for Generalisation in Reinforcement Learning

## Abstract

In multi-task reinforcement learning, agents train on a fixed set of tasks and have to generalise to new ones. Recent work has shown that increased exploration improves this generalisation, but it remains unclear why exactly that is. In this paper, we introduce the concept of *reachability* in multi-task reinforcement learning and show that an initial exploration phase increases the number of reachable tasks the agent is trained on. This, and not the increased exploration, is responsible for the improved generalisation, even to unreachable tasks. Inspired by this, we propose a novel method *Explore-Go* that implements such an exploration phase at the beginning of each episode. Explore-Go only modifies the way experience is collected and can be used with most existing on-policy or off-policy reinforcement learning algorithms. We demonstrate the effectiveness of our method when combined with some popular algorithms and show an increase in generalisation performance across several environments.

## 1 Introduction

Despite major advances in reinforcement learning (RL), it is fairly rare to encounter RL outside of the academic setting. One of the remaining challenges of adopting it in the real world is the ability of an agent to generalise to novel scenarios, that is, those not encountered during training. For example, we do not want a house-cleaning robot to stop working when the owner moves their couch. This is the main research question investigated in the zero-shot policy transfer setting (ZSPT, Kirk et al., 2023). Here the agent trains on several variations of an environment, known as tasks, and must generalise to new ones. This differs from the commonly studied single-task RL setting, in which the agent trains and tests on the same environment instance.

There exists a surprising interaction between ZSPT generalisation and exploration of the training environments. A single-task RL agent must trade off between exploring for better futures and exploiting what it already knows. Once a good enough policy is found, a single-task agent ceases exploration to focus on collecting rewards. In multi-task RL, however, Jiang et al. (2023) have recently demonstrated that more effective exploration, that never stops throughout the entire training process, improves generalisation to unseen tasks.

However, it is not yet entirely clear in *which tasks* we can expect exploration to improve generalisation, nor is it clear *when* to use it to benefit generalisation the most.[1] For example, exploration might help a cleaning robot to know what to do when it is activated at an unusual location in the house. Having seen more of the environment means the robot will be familiar with that area. However, if the owner rearranges some furniture, the previous path to move might be blocked, and it is unclear if and how more exploration would help in this situation.

In this paper, we address these questions by introducing the concept of *reachability* to multi-task RL. We define a task to be reachable if it contains states and rewards that also appear in at least one of the training tasks. Conversely, an unreachable task shares no states and/or rewards with any of the training tasks. In the example above, activating the robot in an unusual location is a reachable task, whereas moving the furniture creates an unreachable one. The key difference between the two is that reachable tasks have states that can be explicitly encountered and optimised during training, whereas

---

[1]Jiang et al. (2023) do provide one possible explanation for why exploration can benefit generalisation, but as we argue in Appendix A.2, this explanation does not cover all scenarios encountered in the ZSPT setting.

unreachable ones do not. However, we argue that training on more reachable tasks results in a form of implicit data augmentation. As data augmentation is frequently shown to improve generalisation in a wide range of settings (Shorten & Khoshgoftaar, 2019; Feng et al., 2021; Zhang et al., 2021a; Miao et al., 2023), we postulate this is responsible for the increase in test performance in unreachable tasks. For example, the above robot might learn to steer around *any* furniture while navigating throughout the house, which can become useful when *some* of it is moved. Our contributions are the following:

- We introduce the concept of reachable and unreachable tasks in reinforcement learning and argue that exploration can be used to increase the number of tasks on which the agent trains. Moreover, we argue that training on these additional reachable tasks can improve generalisation, even to unreachable ones.

- We propose a novel method called *Explore-Go* that can be combined with most existing on-policy or off-policy RL algorithms. It leverages an exploration phase at the beginning of each episode to artificially increase the number of tasks on which the agent trains. We show that Explore-Go can improve generalisation performance to reachable and unreachable tasks when combined with on-policy or off-policy methods.[2]

- We empirically show that generalisation performance is more correlated with the decision *when* the agent explores and how many reachable *tasks* it can solve optimally, rather than *how much* it explores and how many of the reachable *states* it is optimal in.

## 2 BACKGROUND

A Markov decision process (MDP) $\mathcal{M}$ is defined by a 6-tuple $\mathcal{M} = \{S, A, R, T, p_0, \gamma\}$. In this definition, $S$ denotes a set of states called the state space, $A$ a set of actions called the action space, $R : S \times A \to \mathbb{R}$ the reward function, $T : S \times A \to \mathcal{P}(S)$ the transition function where $\mathcal{P}(S)$ denotes the set of probability distributions over states $S$, $p_0 : \mathcal{P}(S)$ the starting state distribution and $\gamma \in [0, 1)$ a discount factor. The goal is to find a policy $\pi : S \to \mathcal{P}(A)$ that maps states to probability distributions over actions in such a way that maximises the expected cumulative discounted reward $\mathbb{E}_\pi[\sum_{t=0}^\infty \gamma^t r_t]$, also called the *return*. The expectation $\mathbb{E}_\pi$ is over the Markov chain $\{s_0, a_0, r_0, s_1, a_1, r_1...\}$ induced by policy $\pi$ when acting in MDP $\mathcal{M}$ (Akshay et al., 2013). An optimal policy $\pi^*$ achieves the highest possible return. The on-policy distribution $\rho^\pi : \mathcal{P}(S)$ of the Markov chain induced by policy $\pi$ in MDP $\mathcal{M}$ defines the proportion of time spent in each state as the number of episodes in $\mathcal{M}$ goes to infinity (Sutton & Barto, 2018).

### 2.1 CONTEXTUAL MARKOV DECISION PROCESS

A contextual MDP (CMDP, Hallak et al., 2015) is a specific type of MDP where the state space $S = S' \times C$ can in principle be factored into an underlying state space $S'$ and a context space $C$, which affects rewards and transitions of the MDP. For a state $s = (s', c) \in S$, the context $c$ behaves differently than the underlying state $s'$ in that it is sampled at the start of an episode (as part of the distribution $p_0$) and remains fixed until the episode ends. The context $c$ can be thought of as the task an agent has to solve and from here on out we will refer to the context as the task.

The zero-shot policy transfer (ZSPT, Kirk et al., 2023) setting for CMDPs $\mathcal{M}|_C$ is defined by a distribution over task space $\mathcal{P}(C)$ and a set of tasks $C^{train}$ and $C^{test}$ sampled from the same distribution $\mathcal{P}(C)$. The goal of the agent is to maximise performance in the testing CMDP $\mathcal{M}|_{C^{test}}$, defined by the CMDP induced by the testing tasks $C^{test}$, but the agent is only allowed to train in the training CMDP $\mathcal{M}|_{C^{train}}$. The learned policy is expected to perform *zero-shot* generalisation for the testing tasks, without any fine-tuning or adaptation period.

## 3 THE INFLUENCE OF REACHABILITY ON GENERALISATION

In general, the task $c$ can influence several aspects of the underlying MDP, like the reward function or dynamics of the environment. As a result, several existing fields of study like multi-goal RL (task

---

[2]We provide code for our experiments at `<redacted for review>`.

influences reward) or sim-to-real transfer (task influences dynamics and/or visual observations) can be framed as special instances of the CMDP framework. To analyse which tasks can generalise to each other, we assume the full state is observed in a representation $s = \phi(s', c)$, such that two tasks that *behave* the same[3] are *represented* the same. This means tasks $c$ only differ in the distribution of their starting states $s_0 \sim p_0(c)$. Many interesting problems are represented in this fashion, including several environments from the popular Procgen, DeepMind Control Suite and Minigrid benchmarks (Cobbe et al., 2020; Tassa et al., 2018; Chevalier-Boisvert et al., 2023).

In this setting, the agent starts a task in a different state but may still *share* states $s_t$ with other tasks later in the episode. For example, if tasks have different starting positions but share the same goal, or if the agent can manipulate the environment to resemble a different task. This is not necessarily always true, though. An example of this is shown in Figure 1a: even if the agent in Task 1 moves to the starting location in Task 2, the background colour will always be different. In this setting, we can refer to tasks $c \in C$ and states $s \in S$ interchangeably, since we can think of any $s$ as a starting state and therefore as a task. From now on, we will refer to a set of tasks $C$ as a set of starting states $S_0$.

### 3.1 REACHABILITY IN MULTI-TASK RL

To argue how exploration can benefit generalisation we introduce the reachability of *tasks*. To do so, we first define the reachability of *states* in a CMDP $\mathcal{M}|_{S_0^{train}}$. The set of reachable states $S_r(\mathcal{M}|_{S_0^{train}})$ (abbreviated with $S_r$ from now on) consists of all states $s_r$ for which there exists a sequence of actions that give a non-zero probability of ending up in $s_r$ when performed in $\mathcal{M}|_{S_0^{train}}$. Put differently, a state $s_r$ is reachable if there exists a policy whose probability of encountering that state during training is non-zero. In complement to reachable states, we define *unreachable* states $s_u$ as states that are not reachable.

Using these definitions, we define (un)reachable tasks as tasks that start in a(n) (un)reachable state. We define two instances of the ZSPT problem as follows:

**Definition 1** (Reachable/Unreachable generalisation). *Reachable/Unreachable generalisation refers to an instance of the ZSPT problem where the start states of the testing environments $S_0^{test}$ are/are-not part of the set of reachable states during training, i.e. $S_0^{test} \subseteq S_r$ or $S_0^{test} \cap S_r = \emptyset$.*

This definition has some interesting implications: due to how reachability is defined, in the reachable generalisation setting all states encountered in the testing CMDP $\mathcal{M}|_{S_0^{test}}$ are also reachable. Note that the reverse does not have to be true: not all reachable states can necessarily be encountered in $\mathcal{M}|_{S_0^{test}}$. Furthermore, we assume in the unreachable generalisation setting that all states encountered in $\mathcal{M}|_{S_0^{test}}$ are also unreachable.[4] Note that this is still considered *in-distribution* generalisation since the starting states for both train and test tasks are sampled from the same distribution.

### 3.2 GENERALISATION TO REACHABLE TASKS

In the single-task setting, the goal is to maximise performance in the MDP $\mathcal{M}$ in which the agent trains. There, it is sufficient to learn an optimal policy in all the states $s \in S$ encountered by this policy in $\mathcal{M}$. This is because acting optimally in all the states encountered by the optimal policy in $\mathcal{M}$ guarantees maximal return in $\mathcal{M}$. Exploration thus only has to facilitate learning the optimal policy on the on-policy distribution $\rho^{\pi^*}$ of $\mathcal{M}$. In fact, once the optimal policy has been found, learning to be optimal anywhere else in $\mathcal{M}$ would be a wasted effort that potentially allocates approximation power to unimportant areas of the state space.

Recent work has shown that this logic does not transfer to the ZSPT problem setting (Jiang et al., 2023). In this setting, the goal is not to maximise performance in the training CMDP $\mathcal{M}|_{S_0^{train}}$, but rather to maximise performance in the testing CMDP $\mathcal{M}|_{S_0^{test}}$. Ideally, the learned policy will be optimal over the on-policy distribution $\rho^{\pi^*}$ in this testing CMDP.

---

[3]Formally: iff for all underlying states $s'$ and actions $a$ the reward and transition models are the same.

[4]This holds for ergodic CMDPs. However, in some non-ergodic CMDPs, it is possible that you can transition into the reachable set $S_r$ after starting in an unreachable state, which we do not consider in this paper.

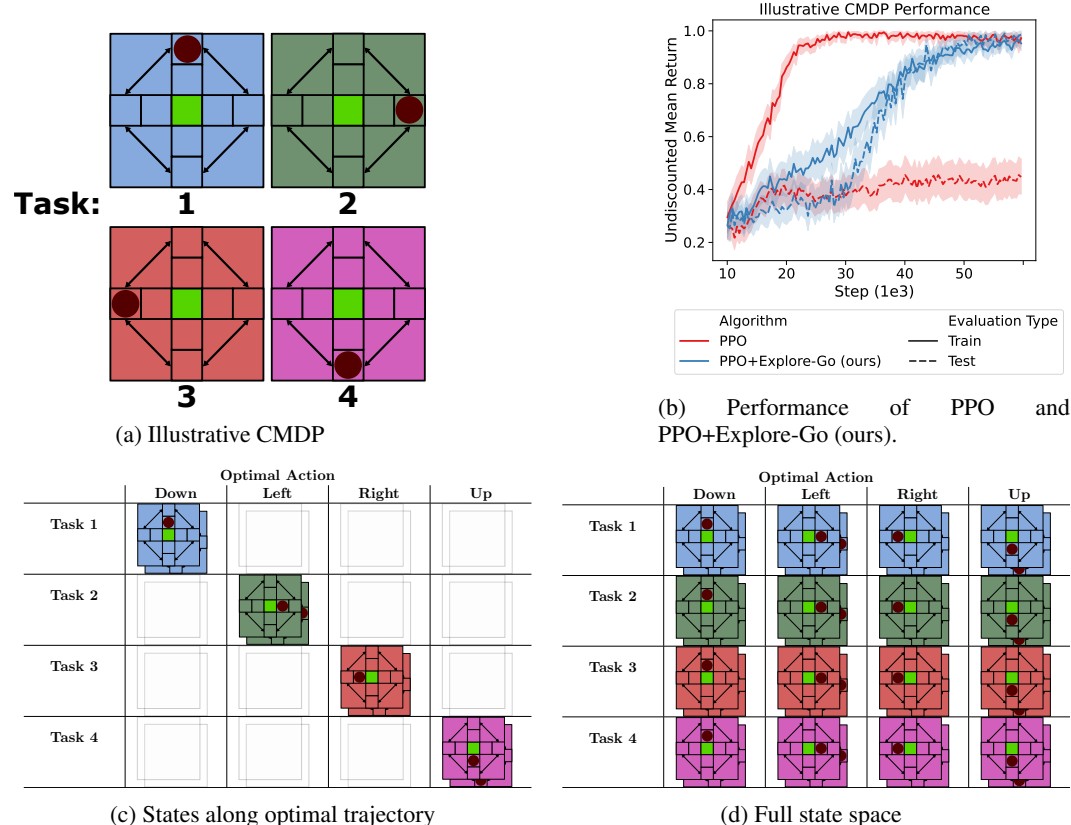

(a) Illustrative CMDP

(b) Performance of PPO and PPO+Explore-Go (ours).

(c) States along optimal trajectory

(d) Full state space

Figure 1: (a) Illustrative CMDP with four training tasks, each with a different background colour and starting position (circle). All tasks share the same goal location (green square in the middle). (b) Performance of a baseline PPO agent and our Explore-Go agent on the CMDP. The agent trains on the tasks in (a) and is tested in tasks with a completely new background colour. Shown are the mean and 95% confidence interval over 100 seeds. Below are (c) the states along the optimal trajectories, and (d) the reachable state space, categorised by their task (rows) and their optimal action (columns).

In general, this testing distribution is unknown. However, in the reachable generalisation setting, the starting states during testing are (by definition) part of the reachable state space $S_r$. So, an agent that learns to act optimally in as many of the reachable states as possible can improve its performance during testing. In fact, if a policy were optimal on all reachable states, it would be guaranteed to 'generalise' to any reachable task (see Appendix B for more detail). In this way, more extensive exploration can help the agent train on more reachable states, which can result in increased 'generalisation' performance. One could argue generalisation is not the best term to use here, since even a policy that completely overfits to the reachable state space $S_r$, for example, a tabular setting, would exhibit perfect 'generalisation'.

### 3.3 GENERALISATION TO UNREACHABLE TASKS

For unreachable generalisation, the states encountered in $\rho^{\pi^*}$ of $\mathcal{M}|_{S_0^{test}}$ are not part of the reachable space $S_r$ of $\mathcal{M}|_{S_0^{train}}$, so it is not obvious on which parts of $S_r$ our agent should train.

To investigate this, we define an example CMDP in Figure 1a. This CMDP consists of a cross-shaped grid world with additional transitions that directly move the agent between adjacent end-points of the cross (e.g., moving right at the end-point of the northern arm of the cross will move you to the eastern arm). The goal for the agent (circle) is to move to the centre of the cross (the green square). There are four training tasks which differ in the starting location of the agent and the colour of the

background. In Figure 1c the states from the optimal trajectories are placed in the table according to what task they are from (row) and what action is optimal (column).

To succeed in the single-task setting (consider just one of our four tasks), an agent only needs to learn to act in the states along the optimal trajectory. Along the optimal trajectories, the colour of the background is perfectly correlated with the optimal action, so a policy trained with a standard RL algorithm will likely overfit to this correlation. As a result, this policy is unlikely to generalise to new *reachable* states (empty cells from the same row/task in Figure 1c), and even less likely to new *unreachable* states with an unseen background colour (a completely new row). We show this empirically in Figure 1b where an agent trained with proximal policy optimisation (PPO, Schulman et al., 2017, red) does not generalise to tasks with a new background colour (see Appendix C.1 for more on this experiment).

Suppose now, we have a policy that has learned over the entire reachable state space (see Figure 1d). This agent is more likely to learn to ignore the background colour, as it no longer correlates with the optimal action. We see this ability to uncover the true relationships and generalise to new colours when using our novel method PPO+Explore-Go (blue in Figure 1b), which effectively trains on all reachable tasks (Explore-Go is further introduced in Section 4).

More generally, we can view the inclusion of additional reachable states (those in Figure 1d which are not in Figure 1c) as a form of data augmentation. For example, the additional states from tasks 2, 3 and 4 in the first column in Figure 1d, can be viewed as simple visual transformations of the state from Task 1 that do not affect the underlying meaning. Data augmentation is commonly used to improve generalisation performance in a wide variety of settings and applications (Shorten & Khoshgoftaar, 2019; Feng et al., 2021; Zhang et al., 2021a; Miao et al., 2023) and is thought to work by reducing overfitting to spurious correlations (Shen et al., 2022), inducing model invariance (Lyle et al., 2020; Chen et al., 2020) and/or regularising training (Bishop, 1995; Lin et al., 2022). Considering the strong evidence of data augmentation's effect on generalisation, we postulate that generalisation to unreachable tasks can be improved by performing data augmentation in the form of training on more reachable tasks.

Note that this data augmentation only works if we know the correct *targets* for the extra samples (columns in Figure 1d). These targets can be optimal actions for policies, or expected returns for (Q-)value functions. If the targets are not correct, the agent might still overfit to a spurious correlation, or worse, learn the wrong function. From the model invariance perspective, not only does training with the incorrect targets not learn the desired invariance, but it explicitly trains to not be invariant. This will likely not improve generalisation and could instead drastically deteriorate it.

Extended exploration (as in Jiang et al., 2023) chooses trajectories that visit more states, but those can sometimes provide poor target estimates. However, as we argue above, training on even a small number of samples with incorrect targets can be harmful. Instead, the expected return is best estimated using rollouts of the current policy. By treating the additional sample as the starting state of a reachable task, we can rely on the RL algorithm to converge to an optimal policy from this state, resulting in accurate targets. Most algorithms, both on- and even off-policy, collect mainly on-policy data towards the end of training. This reduces training on exploratory data with incorrect targets. The next section introduces our novel method Explore-Go, which achieves significantly better generalisation with this approach.

## 4 EXPLORE-GO: TRAINING ON MORE REACHABLE TASKS

As argued in the previous section, training on more reachable tasks is more desirable for generalisation than extended exploration. We propose a novel method *Explore-Go*[5] which effectively trains from more reachable tasks by artificially increasing the diversity of the starting state distribution. It achieves this by introducing an exploration period at the start of each training episode.

Our method is implemented by modifying a fundamental part of most RL algorithms: the collection of rollouts. At the start of every episode, before the agent collects its experiences, Explore-Go

---

[5]The name Explore-Go is a variation of the popular exploration approach Go-Explore (Ecoffet et al., 2021). In Go-Explore the agent teleports at the start of each episode to a novel state and then continuous exploration. In our approach, the agent first explores until it finds a novel state and then goes and solves the original task.

first enters a phase in which it explores the environment by following a *pure exploration* policy. Pure exploration refers to an objective that ignores the rewards $r_t$ the agent encounters and instead focuses purely on exploring new parts of the state space. This pure exploration phase will proceed for $k$ steps. Wherever the pure exploration phase ends will be treated by the agent as the starting state of that episode. This means the rest of the episode continues as it would usually, including any exploration that the agent might normally perform. To add some additional stochasticity to the induced starting state distribution, the length of the pure exploration phase is uniformly sampled between 0 and some fixed value $K$ at the start of every episode. See Algorithm 1 in the appendix for an example of a generic rollout collection protocol modified with Explore-Go.

The basic version of Explore-Go used in this paper does not use the experience collected during the pure exploration phase in any way. In theory, this experience can be used by off-policy methods. However, in Appendix D.1 we show that adding this experience to the replay buffer in deep Q-learning (DQN, Mnih et al., 2015) does not improve performance. However, this experience can be used to train a separate pure exploration agent in parallel to the main agent. In Appendix E we provide the pseudo-code of this version of Explore-Go when combined with PPO.

Note that even though Explore-Go changes the distribution of the training data, it can be combined with both off-policy *and* on-policy reinforcement learning methods. On-policy approaches typically require (primarily) on-policy data for training, distributed along the on-policy state distribution $\rho^{\pi_\theta}(\mathcal{M}|_{S_0^{train}})$ of the current policy $\pi_\theta$. This means they won't work with arbitrary changes to the distribution of training data. However, Explore-Go only changes the distribution of the starting states $S_0^{train}$. So, we can think of Explore-Go as generating on-policy data for a modified MDP that differs only in its starting state distribution. As such, it can be combined with most on-policy approaches.

## 5 EXPERIMENTS

We perform an empirical evaluation of Explore-Go on some environments from two benchmarks: an adaptation of Four Rooms from Minigrid (Chevalier-Boisvert et al., 2023) and Finger Turn and Reacher from the DeepMind Control Suite (DMC, Tassa et al., 2018). These environments can all be explored sufficiently with $\epsilon$-greedy exploration and therefore for the pure exploration policy we simply sample uniformly from the action space (equivalent to setting $\epsilon = 1$). Due to its discrete nature and smaller size, we use the Four Rooms environment to demonstrate the versatility of Explore-Go. This also allows us to enumerate all possible states and tasks and formulate optimal policies and values, which we can use to further analyse our method. We evaluate Explore-Go when combined with several on-policy, off-policy, value-based and/or policy-based RL algorithms: PPO (on-policy, policy-based), DQN (off-policy, value-based) and soft actor-critic (SAC, off-policy, policy-based, Haarnoja et al., 2018).

### 5.1 EXPLORE-GO WITH VARIOUS ALGORITHMS

We use the Four Rooms environment from Minigrid, modified to have a reduced action space, smaller size, and to be fully observable (see Appendix C.2 for more details). The environment consists of a grid-world of four rooms with single-width doorways connecting all of the rooms. The agent starts in one of the rooms and must move to the goal location, which may be in a different room. Tasks differ from each other in the starting location and orientation of the agent, the goal location, and the position of the doorways connecting the four rooms. In our experiments, the agents train on 40 different training tasks and are evaluated on either 120 reachable tasks or 120 unreachable tasks. In this environment, a task is reachable if and only if both the positions of the doorways and the goal location are the same as at least one task in the training set. In Figure 2 we see that Explore-Go improves the testing performance on unreachable tasks when combined with PPO, DQN and SAC, whilst leaving the training performance mostly unaffected. The Explore-Go agent has a maximum of $K = 60$ pure exploration steps at the start of each episode. For more experimental details we refer to Appendix C.2.

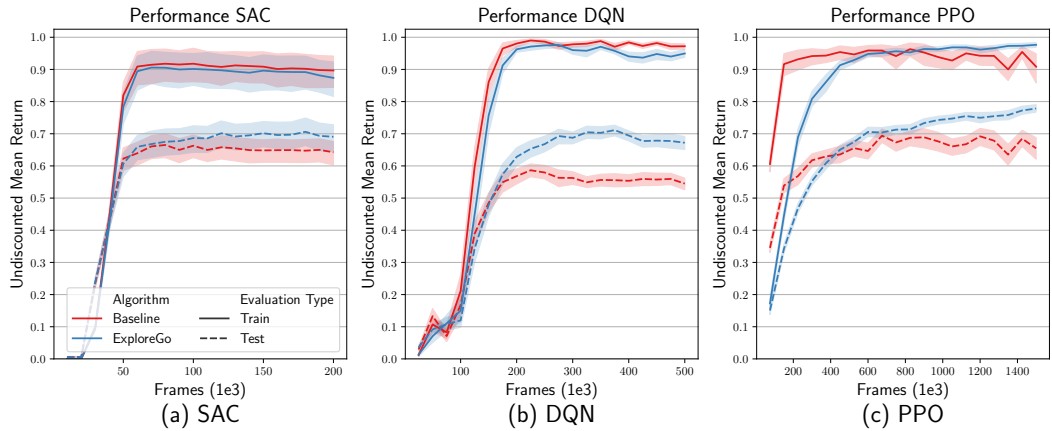

(a) SAC        (b) DQN        (c) PPO

Figure 2: Training and unreachable testing performance of Explore-Go in the Four Rooms environment when combined with (a) SAC, (b) DQN and (c) PPO. Shown are the mean and 95% confidence intervals for 100, 50 and 50 seeds, respectively.

## 5.2 REACHABLE STATES VS REACHABLE TASKS

Our method Explore-Go aims to create additional reachable tasks on which the agent trains. We argue that this, and not simply more continued exploration, will improve generalisation. To investigate this, we compare Explore-Go with an exploration approach that is similar to what is used in Jiang et al. (2023). One of their core algorithmic components is the temporally equalised exploration (TEE) which assigns different fixed exploration coefficients to the parallel workers collecting rollouts.[6] This is necessary because, due to function approximation, the model may lose knowledge acquired through exploration if it does not keep exploring throughout training.

In the following experiment, we analyse the DQN agent from the previous section, which collects rollouts with 10 parallel workers. For the TEE agent, we assign each of the workers a different, fixed value of $\epsilon$ (used in $\epsilon$-greedy exploration). We assign $\epsilon$ according to the relation $\epsilon_i = (\frac{i}{N-1})^\alpha$, where $\epsilon_i$ is the exploration coefficient for worker $i$, $N$ is the total number of workers ($N = 10$ in our case) and $\alpha$ is a coefficient determining a bias towards more exploration ($\alpha < 1$) or less exploration ($\alpha > 1$).

We compare Explore-Go with a baseline DQN agent using TEE with coefficient $\alpha = 0.1$. This was decided by evaluating multiple coefficients $\alpha$ and finding that DQN-TEE with coefficient $\alpha = 0.1$ does the most exploration, and thus acts as an upper bound on the performance achievable with this approach. (see Appendix D.2 for more results with different values of $\alpha$). Figure 3 shows that Explore-Go achieves significantly higher testing performance for both the reachable and unreachable test sets, whilst training performance is largely similar.

In Figure 4 we show that despite discovering a larger fraction of the state-action space (Figure 4a), maintaining higher diversity in the replay buffer (Figures 4b and 4c), and learning the optimal action on a larger fraction of the reachable state space (Figure 4d), TEE generalises *worse* than Explore-Go (as seen in Figure 3). We refer to Appendix C.2 for more details on how these metrics are calculated. This suggests that generalisation is not about *how much* you explore or how many of the reachable *states* you are optimal in, but rather *when* you explore and how many reachable *tasks* you can solve optimally. Our method Explore-Go leverages exploration at the start of every episode to explicitly increase the number of tasks the agent trains on, resulting in consistently higher generalisation performance.

---

[6]Their approach also uses ensembles and distributional RL in conjunction with UCB (Lattimore & Szepesvari, 2017) to explore the environment. We instead use $\epsilon$-greedy since we find it works well in Four Rooms.

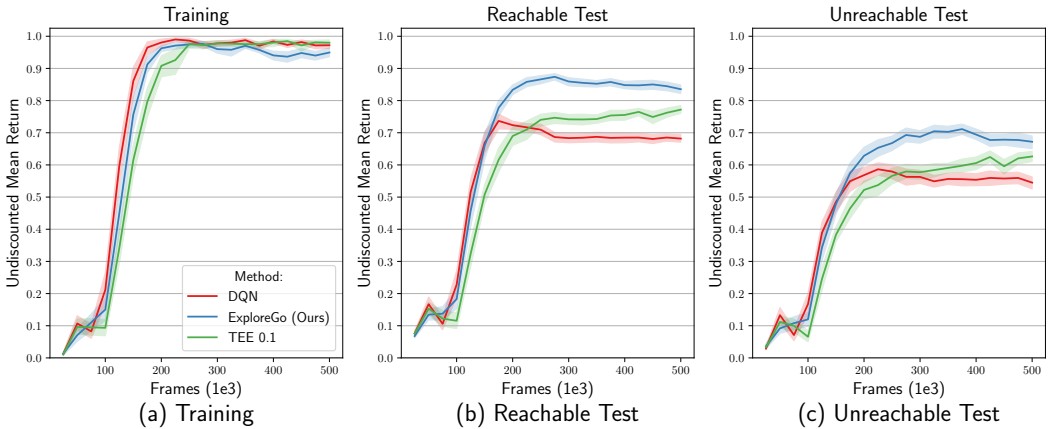

Figure 3: Performance of DQN, DQN+Explore-Go and DQN+TEE with coefficient $\alpha = 0.1$ in Four Rooms on the (a) training set, (b) reachable test set and (c) unreachable test set. Shown are the mean and 95% confidence intervals over 50 seeds.

## 5.3 SCALING UP TO DEEPMIND CONTROL SUITE

To further demonstrate the scalability and generality of our approach we evaluate Explore-Go on some of the continuous control environments from the DeepMind Control Suite. In the DMC environments, at the start of every episode, the initial configuration of the robot body (and in some environments, target location) is randomly generated based on some initial seed. Typically, the DMC benchmark is not used for the ZSPT setting and training is done on the full distribution of tasks (initial configurations). To turn the DMC benchmark into an instance of the ZSPT problem, we define a limited set of seeds (and therefore initial configurations) on which the agents are allowed to train. We then test on the full distribution. Note that only some of the environments test for unreachable generalisation: Reacher, Finger Turn, Manipulator, Stacker, Fish and Swimmer. For the other environments, all tasks are reachable from one another. For more details on these experiments, we refer to Appendix C.3.

In Figure 5 we show the training and testing performance of SAC and Explore-Go on Finger Turn and Reacher. The Explore-Go agent has a maximum of $K = 200$ pure exploration steps at the start of every episode. In the figure, we see it achieves higher test performance whilst leaving training performance largely unaffected. In Appendix D.3 we also show the results for the Cheetah Run and Walker Walk environments. However, there appears to be no significant generalisation gap between

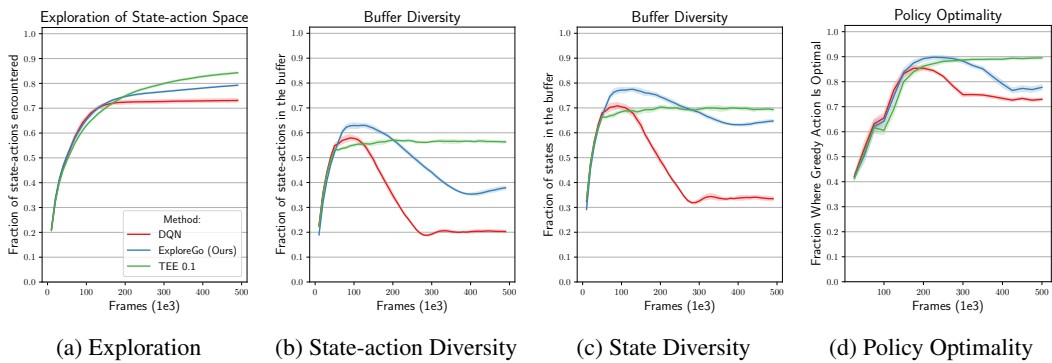

Figure 4: Comparing DQN, DQN+Explore-Go and DQN+TEE with coefficient $\alpha = 0.1$ in Four Rooms for (a) fraction of state-action space explored, (b) fraction of state-action or (c) state space in the buffer and (d) fraction of states where the policy chooses the optimal action. Shown are the mean and 95% confidence intervals over 10 seeds for (a)-(c) and 50 seeds for (d).

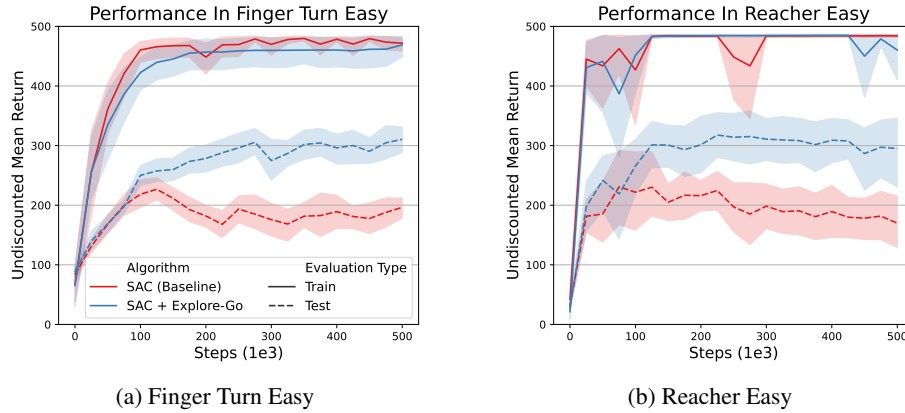

(a) Finger Turn Easy        (b) Reacher Easy

Figure 5: Performance of SAC and Explore-Go on state-based (a) Finger Turn Easy and (b) Reacher Easy. Shown are the mean and 95% confidence intervals over 10 seeds.

training and testing in either environment. Due to this, we focus on the Finger Turn and Reacher environments for our main results.

The experiments above train on the original DMC configuration where the observation an agent receives is a short vector-based state that includes all of the relevant information about the state of the environment. It is also possible to train on DMC with images as observations. Figure 6 shows the performance of Explore-Go on the Finger Turn and Reacher when training on the image-based observations. As a baseline, we use RAD (Laskin et al., 2020) which is SAC with automatic random cropping data augmentation. Figure 6 shows that Explore-Go can also improve generalisation performance on Finger Turn and Reacher when training on image-based observations.

## 6 RELATED WORK

The contextual MDP framework is a very general framework that encompasses many fields in RL that study zero-shot generalisation. Some approaches in this field try to improve generalisation by increasing the variability of the training tasks through domain randomisation (Tobin et al., 2017; Sadeghi & Levine, 2017) or data augmentation (Raileanu et al., 2021; Lee et al., 2020). Others try to explicitly bridge the gap between the training and testing tasks through inductive biases (Kansky et al., 2017; Wang et al., 2021) or regularisation (Cobbe et al., 2019; Tishby & Zaslavsky, 2015). We mention only a small selection of approaches here, for a more comprehensive overview we refer

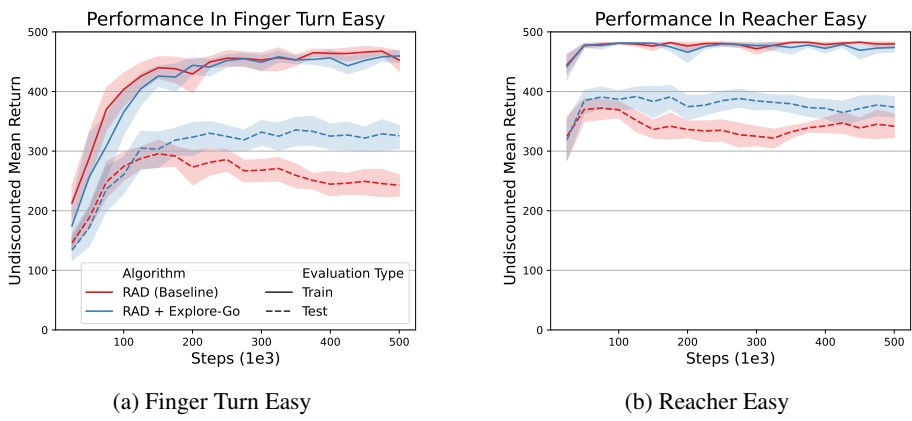

(a) Finger Turn Easy        (b) Reacher Easy

Figure 6: Performance of RAD and Explore-Go on image-based (a) Finger Turn Easy and (b) Reacher Easy. Shown are the mean and 95% confidence intervals over 10 seeds.

to Appendix A.1 or the survey by Kirk et al. (2023). All these approaches use techniques that are not necessarily specific to RL (representation learning, regularisation, etc.). In this work, we instead explore how exploration in RL can be used to improve generalisation.

Next, we discuss related work on exploration in CMDPs. Zisselman et al. (2023) leverage exploration at test time to move the agent towards states where it can confidently solve the task, thereby increasing test time performance. Our work differs in that we leverage exploration during training in order to increase the number of states from which the agent can confidently solve the test tasks. More closely related is work by Jiang et al. (2023), Zhu et al. (2020) and Suau et al. (2024). Jiang et al. (2023) do not make a distinction between reachable and unreachable generalisation and provide intuition which we argue mainly applies to reachable generalisation (see Appendix A.2). Moreover, their novel approach only works for off-policy algorithms, whereas ours can be applied to both off-policy and on-policy methods. Zhu et al. (2020) learn a reset controller that increases the diversity of the agent's start states. However, they only argue (and empirically show) that this benefits reachable generalisation. Suau et al. (2024) introduce the notion of policy confounding in out-of-trajectory generalisation. The issue of policy confounding is complementary to our intuition for unreachable generalisation. However, it is unclear how out-of-trajectory generalisation equates to reachable or unreachable generalisation. Moreover, they do not propose a novel, scalable approach to solve the issue.

## 7  CONCLUSION

Recent work shows that more thorough and prolonged exploration can improve generalisation to unseen tasks in multi-task RL. This effect was explained as a result of encountering the same states in testing as were seen during the additional exploration in training. To understand this phenomenon better, we define the notion of *reachability* of states and tasks. This novel perspective makes it clear the above explanation only applies to *reachable tasks*, whereas unreachable tasks only benefit indirectly from the data augmentation that comes with training on more reachable tasks. It also implies that continuous exploration (as in TEE) is not optimal for multi-task generalisation, as the exploratory episodes find more reachable states, but do not learn the task starting from there.

Instead, we define the novel method *Explore-Go*, which begins each episode with a pure exploration phase, before standard learning is resumed. This results in training on more *reachable tasks*, and thus improves generalisation even to unreachable tasks by data augmentation. We show this empirically in the Four Rooms environment: here TEE explores more states, keeps a more diverse replay buffer, and learns a policy that is optimal in more reachable states than Explore-Go. However, Explore-Go generalises better to both reachable and unreachable test tasks. This suggests that generalisation is not about *how much* you explore or how many of the reachable *states* you are optimal in, but rather *when* you explore and how many reachable *tasks* you can solve optimally.

As an added benefit, Explore-Go only requires a simple modification to the sampling procedure, which can be applied easily to most RL algorithms, both on-policy and off-policy. We demonstrate that the method increases multi-task generalisation in the Four Rooms environment with SAC, DQN and PPO. We also show that Explore-Go scales up to more complex tasks from the DeepMind Control Suite, both on the underlying state and on images of the task. We hope to provide practitioners with a simple modification that can improve the generalisation of their agents significantly.

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

# A    RELATED WORK

## A.1    EXTENDED RELATED WORK

### A.1.1    GENERALISATION IN CMDPS

The contextual MDP framework is a very general framework that encompasses many fields in RL that study zero-shot generalisation. For example, the *sim-to-real* setting often encountered in robotics is a special case of the ZSPT setting for CMDPs (Kirk et al., 2023). An approach used to improve generalisation in the sim-to-real setting is domain randomisation (Tobin et al., 2017; Sadeghi & Levine, 2017; Peng et al., 2018), where the task distribution during training is explicitly increased in order to increase the probability of encompassing the testing tasks in the training distribution. This differs from our work in that we don't explicitly generate more (unreachable) tasks. However, our work could be viewed as implicitly generating more reachable tasks through increased exploration. Another approach that increases the task distribution is data augmentation (Raileanu et al., 2021; Lee et al., 2020; Zhou et al., 2021). These approaches work by applying a set of given transformations to the states with the prior knowledge that these transformations leave the output (policy or value function) invariant. In this paper, we argue that our approach implicitly induces a form of invariant data augmentation on the states. However, this differs from the other work cited here in that we don't explicitly apply transformations to our states, nor do we require prior knowledge on which transformations leave the policy invariant.

So far we have mentioned some approaches that increase the number and variability of the training tasks. Other approaches instead try to explicitly bridge the gap between the training and testing tasks. For example, some use inductive biases to encourage learning generalisable functions (Zambaldi et al., 2018; 2019; Kansky et al., 2017; Wang et al., 2021; Tang et al., 2020; Tang & Ha, 2021). Others use regularisation techniques from supervised learning to boost generalisation performance (Cobbe et al., 2019; Tishby & Zaslavsky, 2015; Igl et al., 2019; Lu et al., 2020; Eysenbach et al., 2021). We mention only a selection of approaches here, for a more comprehensive overview we refer to the survey by Kirk et al. (2023).

All the approaches above use techniques that are not necessarily specific to RL (representation learning, regularisation, etc.). In this work, we instead explore how exploration in RL can be used to improve generalisation.

### A.1.2    EXPLORATION IN CMDPS

There have been numerous methods of exploration designed specifically for or that have shown promising performance on CMDPs. Some approaches train additional adversarial agents to help with exploration (Flet-Berliac et al., 2021; Campero et al., 2021; Fickinger et al., 2021). Others try to exploit actions that significantly impact the environment (Seurin et al., 2021; Parisi et al., 2021) or that cause a significant change in some metric (Raileanu & Rocktäschel, 2020; Zhang et al., 2021c;b; Ramesh et al., 2022). More recently, some approaches have been developed that try to generalise episodic state visitation counts to continuous spaces (Jo et al., 2022; Henaff et al., 2022) and several studies have shown the importance of this for exploration in CMDPs (Wang et al., 2023; Henaff et al., 2023). All these methods focus on trading off exploration and exploitation to achieve maximal performance in the training tasks as fast and efficiently as possible. However, in this paper, we examine the exploration-exploitation trade-off to maximise generalisation performance in testing tasks.

In Zisselman et al. (2023), the authors leverage exploration at test time to move the agent towards states where it can confidently solve the task, thereby increasing test time performance. Our work differs in that we leverage exploration during training time to increase the number of states from which the agent can confidently solve the test tasks. Closest to our work is Jiang et al. (2023), Zhu et al. (2020) and Suau et al. (2024). Jiang et al. (2023) don't make a distinction between reachable and unreachable generalisation and provide intuition which we argue mainly applies to reachable generalisation (see Appendix A.2). Moreover, their novel approach only works for off-policy algorithms, whereas ours could be applied to both off-policy and on-policy methods. In Zhu et al. (2020), the authors learn a reset controller that increases the diversity of the agent's start states. However, they only argue (and empirically show) that this benefits reachable generalisation. The concurrent work in Suau et al. (2024) introduces the notion of policy confounding in out-of-

trajectory generalisation. The issue of policy confounding is complementary to our intuition for unreachable generalisation. However, it is unclear how out-of-trajectory generalisation equates to reachable or unreachable generalisation. Moreover, they do not propose a novel, scalable approach to solve the issue.

## A.2 DISCUSSION ON RELATED WORK

Jiang et al. (2023) argue that generalisation in RL extends beyond representation learning. They do so with an example in a tabular grid-world environment. In the environment they describe the agent during training always starts in the top left corner of the grid, and the goal is always in the top right corner. During testing the agent starts in a different position in the grid-world (in their example, the lower left corner). This is according to our definition an example of a reachable task. They then argue (in the way we described in Section 3.2) that more exploration can improve generalisation to these tasks.

They extend their intuition to non-tabular CMDPs by arguing that in certain cases two states that are unreachable from each other, can nonetheless inside a neural network map to similar representations. As a result, even though a state in the input space is unreachable, it can be mapped to something reachable in the latent representational space and therefore the reachable generalisation arguments apply again. For this reason, the generalisation benefits from more exploration can go beyond representation learning.

Relating it to the illustrative example we provide in Figure 1, we argue this intuition considers the generalisation benefits one might obtain from learning to act optimally in more abstracted states. For example, in Jiang et al. (2023)'s grid-world the lower states would have normally unseen values, which is represented by increasing the number of columns on which we train in Figure 1c and 1d. However, in Section 3.2 we argue that specifically unreachable generalisation can benefit as well from training on more states belonging to the same abstracted states (represented by increasing the number of rows on which we train in Figure 1c and 1d). Training on more of these states could encourage the agent to learn representations that map different unreachable states to the same latent representation (or equivalently, abstracted states). As such, we argue the generalisation benefits from more exploration can in part be attributed to an implicit form of representation learning.

## B  GENERALISATION TO REACHABLE TASKS

In this section, we elaborate on why a policy that is optimal in all reachable states, is guaranteed to perform well when testing on reachable tasks. As a first step, we point out a corollary of definition 1 about reachable states:

**Corollary 0.1.** *Any state $s'$ that is reachable from a state $s \in S_r(\mathcal{M}|_{S_0^{train}})$ in the reachable set, has to be itself in the reachable set: $s' \in S_r(\mathcal{M}|_{S_0^{train}})$.*

Why this is the case is clear to see with the definition of reachability in terms of sequences of actions: concatenate the sequence of actions with a non-zero probability of ending up in $s$ with the sequence of actions with a non-zero probability of ending up in $s'$ when starting from $s$. This will result in a sequence of actions with a non-zero probability of ending up in $s'$. In short, this corollary states that you cannot leave the reachable set $S_r(\mathcal{M}|_{S_0^{train}})$ through interaction with the environment.

From this logically follows the following corollary:

**Corollary 0.2.** *An optimal policy $\pi$ that achieves maximal return from any state in the reachable state space $S_r(\mathcal{M}|_{S_0^{train}})$, will have optimal performance in the reachable generalisation setting.*

Recall that performance in a ZSPT problem is defined as the performance in the testing MDP $\mathcal{M}|_{S_0^{test}}$, which in the case of reachable generalisation, has a state space that consists only of reachable states (due to Corollary 0.1). It follows naturally that a policy that is optimal on the entire reachable state space $S_r(\mathcal{M}|_{S_0^{train}})$ also has to be optimal in $\mathcal{M}|_{S_0^{test}}$.

## C  EXPERIMENTAL DETAILS

### C.1  ILLUSTRATIVE CMDP

Training is done on the four tasks in Figure 1a and unreachable generalisation is evaluated on new tasks with a completely different background colour. For pure exploration, we sample uniformly random actions at each timestep ($\epsilon$-greedy with $\epsilon = 1$). We compare Explore-Go to a baseline using regular PPO. In Figure 1b we can see that the PPO baseline achieves approximately optimal training performance but is not consistently able to generalise to the unreachable tasks with a different background colour. PPO trains mostly on on-policy data, so when the policy converges to the optimal policy on the training tasks it trains almost exclusively on the on-policy states in Figure 1c. As we hypothesise, this likely causes the agent to overfit to the background colour, which will hurt its generalisation capabilities to unreachable states with an unseen background colour. On the other hand, Explore-Go maintains state diversity by performing pure exploration steps at the start of every episode. As such, the state distribution on which it trains resembles the distribution from Figure 1d. As we can see in Figure 1b, Explore-Go learns slower, but in the end achieves similar training performance to PPO and performs significantly better in the unreachable test tasks. We speculate this is due to the increased diversity of the state tasks on which it trains.

ENVIRONMENT DETAILS

The training tasks for the illustrative CMDP are the ones depicted in Figure 1a. The unreachable testing tasks consist of 4 tasks with the same starting positions as found in the training tasks (the endpoint of the arms) but with a white background colour. The states the agent observes are structured as RGB images with shape $(3, 5, 5)$. The entire $5 \times 5$ grid is encoded with the background colour of the particular task, except for the goal position (at $(2, 2)$) which is dark green ($(0,0.5,0)$ in RGB) and the agent (wherever it is located at that time) which is dark red ($(0.5,0,0)$ in RGB). The specific background colours are the following:

- **Training task 1:** $(0,0,1)$

- **Training task 2:** $(0,1,0)$

- **Training task 3:** $(1,0,0)$

- **Training task 4:** $(1,0,1)$

- **Testing tasks:** $(1,1,1)$

Moving into a wall of the cross will leave the agent position unchanged, except for the additional transitions between the cross endpoints. Moving into the goal position (middle of the cross) will terminate the episode and give a reward of 1. All other transitions give a reward of 0. The agent is timed out after 20 steps.

IMPLEMENTATION DETAILS

For PPO we used the implementation by Moon et al. (2022) which we adapted for PPO + Explore-Go. The hyperparameters for both PPO and PPO + Explore-Go can be found in Table 1. The only additional hyperparameter that Explore-Go uses is the maximal number of pure exploration steps $K$, which we choose to be $K = 8$. Both algorithms use network architectures that flatten the $(3, 5, 5)$ observation and feed it through a fully connected network with a ReLU activation function. The hidden dimensions for both the actor and critic are $[128, 64, 32]$ followed by an output layer of size $[1]$ for the critic and size $[|A|]$ for the actor. The output of the actor is used as logits in a categorical distribution over the actions.

Table 1: Hyper-parameters used for the illustrative CMDP experiment

| Illustrative | |
|---|---|
| **Hyper-parameter** | **Value** |
| Total timesteps | 50 000 |
| Vectorised environments | 4 |
| **PPO** | |
| timesteps per rollout | 10 |
| epochs per rollout | 3 |
| minibatches per epoch | 8 |
| Discount factor $\gamma$ | 0.9 |
| GAE smoothing parameter ($\lambda$) | 0.95 |
| Entropy bonus | 0.01 |
| PPO clip range ($\epsilon$) | 0.2 |
| Reward normalisation? | No |
| Max. gradient norm | .5 |
| Shared actor and critic networks | No |
| **Adam** | |
| Learning rate | $1 \times 10^{-4}$ |
| Epsilon | $1 \times 10^{-5}$ |

## C.2 FOUR ROOMS

In all of our Four Rooms experiments, we will train on 40 different training tasks and test on either a reachable or unreachable task set of size 120. The 40 training tasks differ in the agent location, agent direction, goal location and the location of the doorways (see Figure 7 for some example tasks in Four Rooms).

In this environment, reachability is regulated through variations in the goal location and location of the doorways. If two states share their doorways and goal location, then they are both reachable from one another. Conversely, if two states differ in either the doorways or goal location, they are unreachable. The reachable task set is constructed by taking every training task and changing only the agent location and agent direction (keeping the location of the doorways and goal location the same). This is repeated four times to generate a total number of reachable tasks of $4 \times 40 = 120$. For the unreachable task set, we take 40 different configurations of the doorways that all differ from the ones in the training task. For each of those 40 different doorway configurations, we generate four new goal locations, agent locations and agent directions. This also generates a total of $4 \times 40 = 120$ unreachable tasks.

### ENVIRONMENT DETAILS

The Four Rooms grid world used in our experiments is adapted from the Minigrid benchmark (Chevalier-Boisvert et al., 2023) and differs in certain ways from the default Minigrid configuration. For one, the action space is reduced from the default seven actions (turn left, turn right, move forward, pick up an object, drop an object, toggle/activate an object, end episode) to just the first three actions (turn left, turn right, move forward). Also, the reward function is changed slightly to reward 1 for successfully reaching the goal and 0 otherwise (as opposed to the $1 - 0.9 * (\frac{\text{step count}}{\text{max steps}})$ given upon success by the default Minigrid environment). Additionally, the size of the environment is reduced from the default 19 ($8 \times 8$ rooms) to 9 ($3 \times 3$ rooms).

Furthermore, the observation space is made fully observable and customised. Our agent receives a $4 \times 9 \times 9$ tensor that is centred around the agent's current location. The four binary-encoded channels contain the following information:

- **Channel 0:** The location of the agent (always in the centre).
- **Channel 1:** The hypothetical location where the agent would move to given the current direction it's facing (and ignoring any collisions with walls).

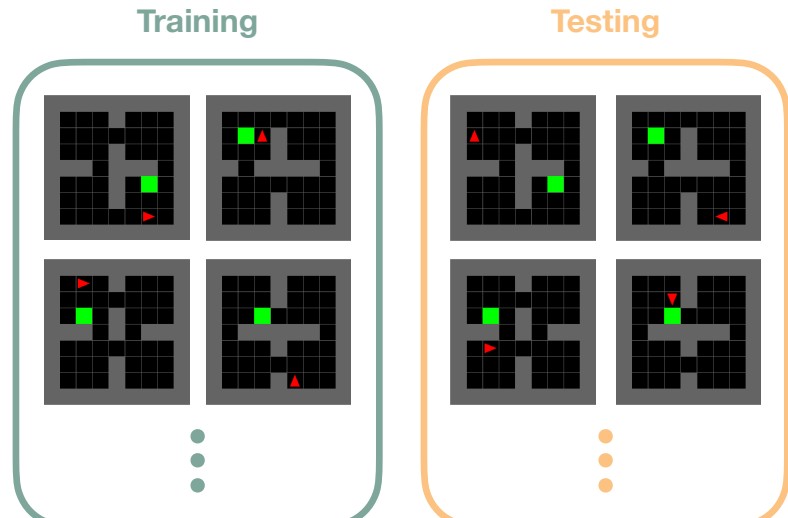

Figure 7: Some example tasks in the Four Rooms environment for reachable generalisation. For unreachable generalisation both the goal and doorway locations would be different in testing.

- **Channel 2:** The location of the walls.

- **Channel 3:** The location of the goal.

The implementation of Four Rooms is also customised to allow for more control over the factors of variation (topology, agent location, agent direction, goal location) during the generation of a task. This acts functionally the same as the `ReseedWrapper` from Minigrid except that it allows for more control and therefore easier design and construction of the training and testing sets. The code for our Four Room implementation can be found at `<redacted for review>`.

EXPLORE-GO WITH DQN, PPO AND SAC

For the DQN, PPO and SAC experiments, we take the implementations from the Stable-Baselines3 (Raffin et al., 2021) repository and add Explore-Go to them (see code at `<redacted for review>`). We adapt the SAC implementation to work with discrete action space. For the DQN implementation, we also add support for double Q learning (van Hasselt et al., 2015). For all experiments, the network architecture consists of three convolutional layers (see parameters in Table 2) followed by some fully connected layers with ReLU activation functions (except for the last layer). The number and width of the fully connected layers depend on the algorithm used. For DQN we have three fully connected layers with hidden dimensions $[512, 128, 64]$. For PPO we have two times three fully connected layers (one for the actor and one for the critic) with hidden dimensions $[512, 128, 64]$. For SAC we have the same but with hidden dimensions $[512, 256, 256]$. A full list of parameters can be found in Table 3 for DQN, Table 4 for PPO and Table 5 for SAC.

The hyperparameter $K$ for Explore-Go that determines the maximum number of steps is chosen by visually inspecting a random agent walking in the Four Rooms environment. The idea behind the process is that we rather have $K$ too big (interactions with the environment wasted), than too small (doesn't find diverse new starting positions). So we choose $K = 60$ for the Four Rooms environment since we find that an average of 30 steps is enough for the agent to randomly explore a decent proportion of the environment.

EXPLORE-GO, DQN AND TEE

For the experiments comparing Explore-Go with DQN and TEE, we use the same hyperparameters as for the other DQN experiments. For the TEE approach, we use a coefficient of $\alpha = 0.1$. For the results with different values of TEE coefficient, we refer to Appendix D.2.

When comparing Explore-Go, DQN and TEE we introduce four new metrics. The first measures the fraction of state-action space that is explored (Figure 4a). This is calculated by enumerating all possible state-actions in the reachable state space and keeping track of which ones are encountered at some point during training. This measures how effective the exploration approach is (a higher fraction means the agent explored more states). The second and third metrics measure the diversity present in the replay buffer throughout training (Figures 4b and 4c). They do so, again, by enumerating all possible state-actions (Figure 4b) or states (Figure 4c) in the reachable space and checking which ones are present in the buffer at that time. The last metric measures how optimal the agent is over the entire reachable space (Figure 4d). It measures this by enumerating all possible states in the reachable space and checking for which ones the agent chooses an action that is optimal (there can be multiple).

Table 2: Hyper-parameters for the CNN part in the Four Rooms experiment

| CNN | |
| --- | --- |
| Kernel size | 3 |
| Stride | 1 |
| Padding | 1 |
| Padding mode | Circular |
| Channels | 32 |

Table 3: Hyper-parameters for Four Rooms DQN

| **Four Rooms DQN** | |
| --- | --- |
| **Hyper-parameter** | **Value** |
| Total timesteps | 500 000 |
| Vectorised environments | 10 |
| Buffer size | 50 000 |
| Batch size | 256 |
| Discount factor $\gamma$ | 0.99 |
| Max. gradient norm | 1 |
| Gradient steps | 1 |
| Train frequency (steps) | 10 |
| Target update interval (steps) | 10 |
| Target soft update coefficient $\tau$ | 0.01 |
| Exploration initial $\epsilon$ | 1 |
| Exploration final $\epsilon$ | 0.01 |
| Exploration fraction $\epsilon$ | 0.5 |
| **Adam** | |
| Learning rate | $1 \times 10^{-4}$ |
| Weight decay | $1 \times 10^{-5}$ |

Table 4: Hyper-parameters for Four Rooms PPO

| Four Rooms PPO | |
|---|---|
| **Hyper-parameter** | **Value** |
| Total timesteps | 1 500 000 |
| Vectorised environments | 10 |
| Batch size | 64 |
| Discount factor $\gamma$ | 0.99 |
| Max. gradient norm | 0.5 |
| # of epochs | 10 |
| # steps collected per rollout | 5 120 |
| Entropy coeff | 0.0 |
| Value function coeff | 0.5 |
| GAE coeff $\lambda$ | 0.95 |
| Share feature extractor | True |
| Clip range | 0.2 |
| **Adam** | |
| Learning rate | $1 \times 10^{-4}$ |

Table 5: Hyper-parameters for Four Rooms SAC

| Four Rooms SAC | |
|---|---|
| **Hyper-parameter** | **Value** |
| Total timesteps | 300 000 |
| Vectorised environments | 10 |
| Buffer size | 200 000 |
| Batch size | 256 |
| Discount factor $\gamma$ | 0.99 |
| Max. gradient norm | 1 |
| Gradient steps | 10 |
| Train frequency (steps) | 10 |
| Target update interval (steps) | 10 |
| Target soft update coefficient $\tau$ | 0.005 |
| Warmup phase | 20 000 |
| Share feature extractor | False |
| Target entropy | auto |
| Entropy coeff | auto |
| **Adam** | |
| Learning rate | $5 \times 10^{-4}$ |

## C.3 DeepMind Control Suite

For the DeepMind Control Suite we adapt the environment so that at the start of each episode the initial configuration of the robot body and target location are drawn based on a given list of random seeds. This allows us to control the task space of the environment so that we can define a limited set of tasks on which the agent is allowed to train. To compute mean performance and confidence intervals we average all our DMC experiments over 10 seeds for the agent. Each agent seed trains on its own set of training tasks. For a training set of size $N$, agent $i$ gets to train on tasks generated with seeds $\{i*N, i*N+1, ..., i*N+N-1\}$. Testing is always done on 100 episodes from the full distribution. For the state-based experiments we train on $N = 5$ training tasks and for the image-based experiments, we train on $N = 30$ training tasks. The code can be found at `<redacted for review>`.

The standard DMC benchmark has no terminal states and instead has a fixed episode length of 1000 after which the agent times out. However, for the Finger Turn and Reacher environments, an episode length of 1000 is unnecessarily long. For these two environments, the goal is to position the robot body in such a way that some designated part is located at a target location. Once it successfully reaches this target location, the optimal policy is to do nothing. This means that in many of the Finger Turn and Reacher episodes, the agent only moves in the first 100 or so steps and then does nothing for 900 more. To simplify the training on these environments a bit we instead shorten the episode length to 500.

For the state-based experiments, we use the Explore-Go and SAC implementation adapted from Stable-Baselines3 (Raffin et al., 2021). Most of the hyperparameters for SAC are taken from (Zhu et al., 2020), but a full list can be found in Table 6. For the image-based experiments, we add Explore-Go to the RAD implementation from (Hansen & Wang, 2021) and use the hyperparameters from (Laskin et al., 2020). For all DMC experiments, we use a maximum pure exploration duration $K = 200$. We judged this to be high enough to generate diverse states in most environments.

Table 6: Hyper-parameters for Four Rooms SAC

| DMC SAC | |
| --- | --- |
| **Hyper-parameter** | **Value** |
| Total timesteps | 500 000 |
| Vectorised environments | 1 |
| Buffer size | 100 000 |
| Batch size | 128 |
| Discount factor $\gamma$ | 0.99 |
| Gradient steps | 1 |
| Train frequency (steps) | 1 |
| Target update interval (steps) | 1 |
| Target soft update coefficient $\tau$ | 0.005 |
| Warmup phase | 10 000 |
| Share feature extractor | False |
| # of layers | 2 |
| Layer size | 256 |
| Target entropy | auto |
| Entropy coeff | auto |
| **Adam** | |
| Learning rate | $1 \times 10^{-3}$ |

# D  ADDITIONAL EXPERIMENTS

## D.1  ADDING PURE EXPLORATION EXPERIENCE TO THE BUFFER

In Figure 8 we show an ablation of Explore-Go where we also add all the pure exploration experience to the replay buffer (Explore-Go with PE, green). It shows that adding this experience to the buffer makes the performance of Explore-Go worse. This could be due to the highly off-policy nature of the pure exploration data.

## D.2  TEE WITH DIFFERENT COEFFICIENTS $\alpha$

TEE has an additional hyperparameter $\alpha$ that determines how much the individual rollout workers are biased towards exploration ($\alpha < 1$) or exploitation ($\alpha > 1$). Figure 9 shows different values of $epsilon_i$ for different values of $\alpha$. Figure 10 shows the training and testing performance and Figure 11 the exploration effectiveness, buffer diversity and policy optimality for the various values of $\alpha$.

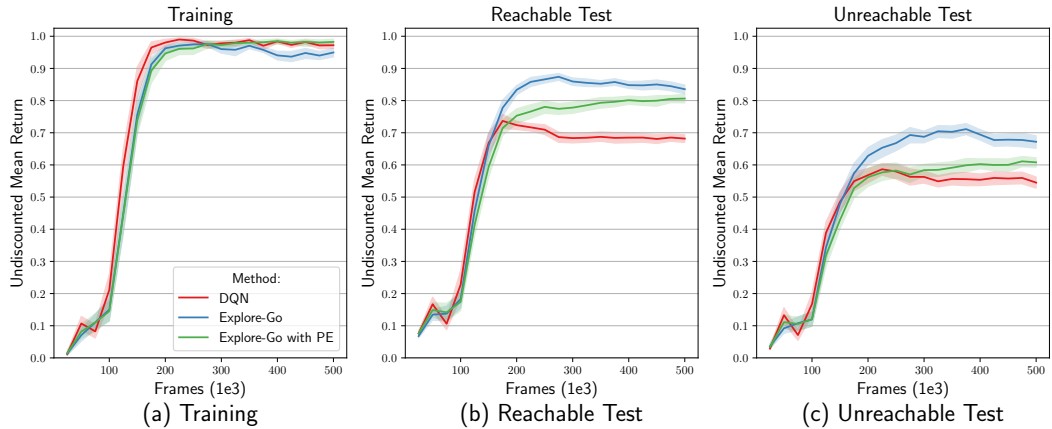

Figure 8: Performance of DQN, DQN+Explore-Go and DQN+Explore-Go where the pure exploration is also added to the replay buffer. Performance is in the Four Rooms environment on the (a) training set, (b) reachable test set and (c) unreachable test set. Shown are the mean and 95% confidence intervals over 50 seeds.

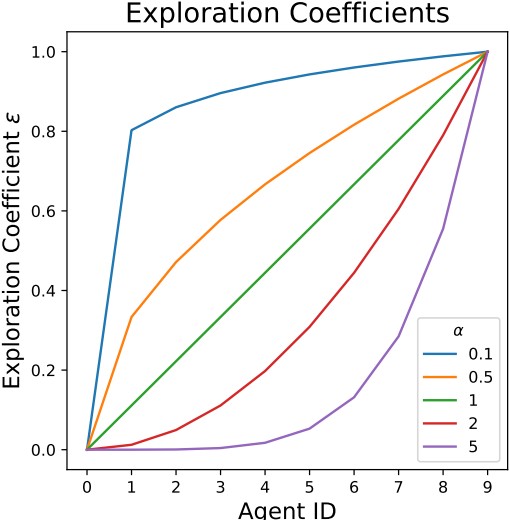

Figure 9: Exploration coefficients $\epsilon_i$ for 10 rollout workers for different values of $\alpha$.

### D.3    CHEETAH RUN AND WALKER WALK

Here we show the results for Cheetah Run and Walker Walk in Figure 12. We use the same hyperparameters as for the other DMC experiments, except we change the episode length back to the original 1000 steps. For both environments we train on task sets of size $N = 5$. In the figure, we can see that for both Cheetah Run and Walker Walk, there is effectively no generalisation gap between training and testing (the solid and dotted lines mostly overlap). This means these environments are not ideal for testing generalisation performance.

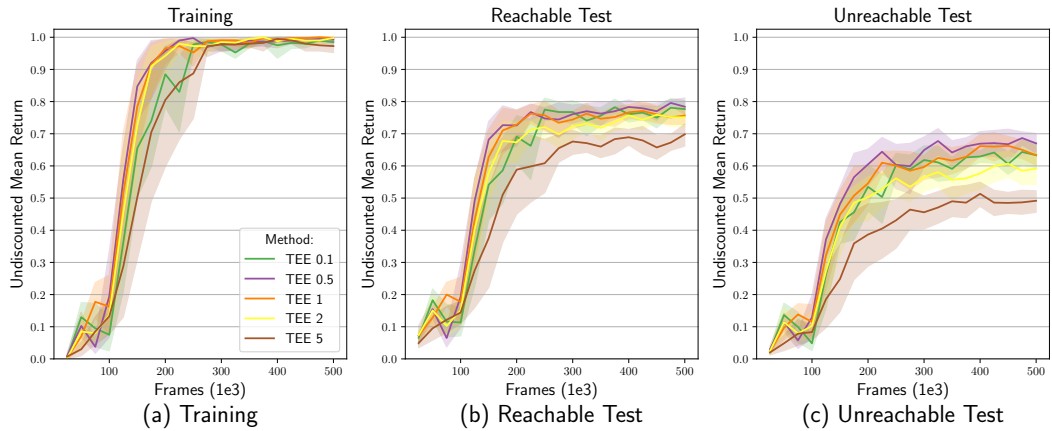

Figure 10: Performance of DQN+TEE with coefficients $\alpha = [0.1, 0.5, 1, 2, 5]$ in Four Rooms on the (a) training set, (b) reachable test set and (c) unreachable test set. Shown are the mean and 95% confidence intervals over 10 seeds.

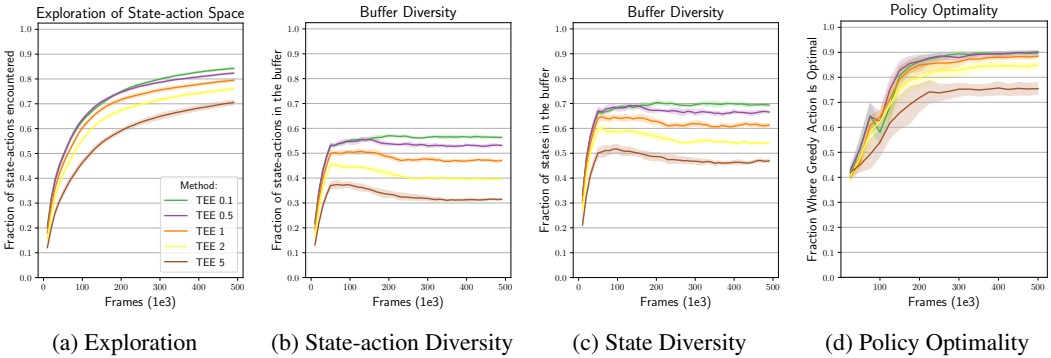

Figure 11: Comparing DQN+TEE with coefficients $\alpha = [0.1, 0.5, 1, 2, 5]$ in Four Rooms for (a) fraction of state-action space explored, (b) fraction of state-action or (c) state space in the buffer and (d) fraction of states where the policy chooses the optimal action. Shown are the mean and 95% confidence intervals over 10 seeds for (a)-(c) and 50 seeds for (d).

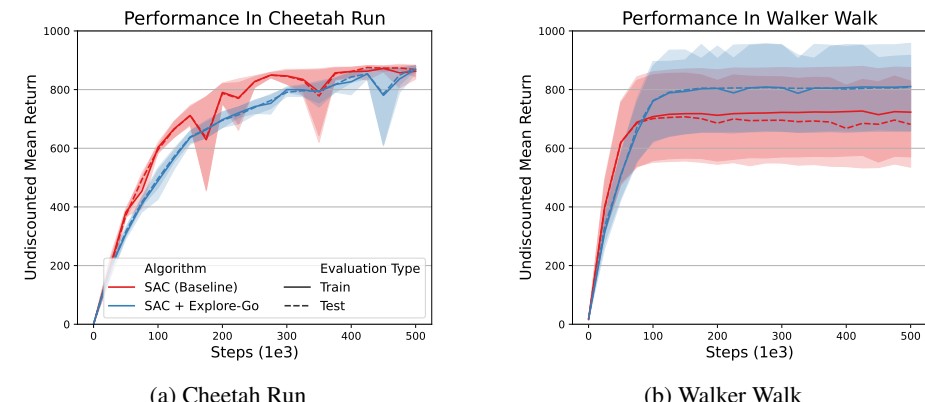

(a) Cheetah Run                                  (b) Walker Walk

Figure 12: Performance of SAC and Explore-Go on state-based (a) Cheetah Run and (b) Walker Walk. Shown are the mean and 95% confidence intervals over 10 seeds.

## E    PSEUDO-CODE

---

**Algorithm 1:** Generic CollectRollouts + Explore-Go

---

**Input:** number of steps to collect $N$, pure exploration policy $\pi_{PE}$, max number of pure
       exploration steps $K$
$k \leftarrow Uniform(0, K)$;
$\mathcal{D}_{rollout} \leftarrow \{\}$;
$num\_steps\_collected \leftarrow 0$;
**while** $num\_steps\_collected < N$ **do**
    **if** $episode\_step < k$ **then**
        Sample transition $t$ using $\pi_{PE}$;
    **else**
        Sample transition $t$;
        Add $t$ to $\mathcal{D}_{rollout}$;
        $num\_steps\_collected$ += 1;
    **end if**
    $episode\_step$ += 1;
    **if** $end\ of\ episode$ **then**
        $k \leftarrow Uniform(0, K)$;
        $episode\_step \leftarrow 0$;
        Reset environment;
    **end if**
**end**
Return $\mathcal{D}_{rollout}$;

---

Figure 13: An example of pseudo-code for Explore-Go combined with a generic rollout collection function found in some form in most RL algorithms.

---

**Algorithm 2:** PPO + Explore-Go

---

**Input:** PPO agent $PPO$, pure exploration agent $PE$, max number of pure exploration steps $K$

$k \leftarrow Uniform(0, K)$;

$i \leftarrow 0$                                    ▷ Counts steps within an episode;

**for** $iteration = 0, 1, 2, ...$ **do**

  $\mathcal{D}_{PPO} \leftarrow \{\}$;

  $\mathcal{D}_{PE} \leftarrow \{\}$;

  **for** $step = 0, 1, 2, ..., T$ **do**

    **if** $i < k$ **then**

      Sample transition $t$ by running $PE$;

      Add $t$ to $\mathcal{D}_{PE}$;

    **else**

      Sample transition $t$ by running $PPO$;

      Add $t$ to $\mathcal{D}_{PPO}$;

    **end if**

    $i \leftarrow i + 1$;

    **if** *end of episode* **then**

      $k \leftarrow Uniform(0, K)$;

      $i \leftarrow 0$;

      Reset environment;

    **end if**

  **end**

  Update $PPO$ with trajectories $\mathcal{D}_{PPO}$;

  (Optional) Update $PE$ with trajectories $\mathcal{D}_{PE}$;

**end**

---

Figure 14: An example of pseudo-code for Explore-Go combined with an on-policy method PPO.