# OpenReview forum: "Training on more Reachable Tasks for Generalisation in Reinforcement Learning"
_ICLR.cc/2025/Conference — ICLR 2025 Conference Withdrawn Submission_

### Official Review · Reviewer_ghi5 · 2024-10-26

**Soundness:** 2
**Presentation:** 2
**Contribution:** 2
**Rating:** 3
**Confidence:** 3

**Summary:**

This paper investigates the impact of exploration on generalization in multi-task reinforcement learning, particularly focusing on the distinction between reachable and unreachable tasks. The authors aim to understand why exploration improves generalization and propose a novel method to enhance it. The authors propose “Explore-Go” which implements an exploration phase at the beginning of each episode. Authors suggest that performance is more correlated with the decision when the agent explores and how many reachable tasks it can solve optimally, rather than how much it explores and how many of the reachable states it is optimal in.

**Strengths:**

-The concept of reachable tasks is interesting

-Training on a wider range of reachable tasks, facilitated by the initial exploration phase in Explore-Go, acts as data augmentation, leading to better generalization.

-Explore-Go consistently improves generalization performance on both reachable and unreachable test tasks in evaluations

**Weaknesses:**

• In theory, if the agent explores infinitely then the optimal value function is reached. So increased exploration should be useful unless comparisons are not properly tuned/ trained.

• In the real world, knowing reachable states and reachable tasks (without T) is going to be very difficult, making this a strong bottleneck to using this approach.

• The need for optimal policy from reachable states (or knowing correct targets) would destroy the purpose of learning.

• Keeping goals and obstacles the same across tasks would ensure the state space is the same, leading to less generalization. This only addresses the problem of variable initial location which makes it less interesting.

• Comparison with SoTA ZSPT, such as [1], [2], etc., is absent.

[1] Roy, Josh, and George Konidaris. "Zero-Shot Policy Transfer with Disentangled Attention."

[2] Higgins, Irina, et al. "Darla: Improving zero-shot transfer in reinforcement learning." International Conference on Machine Learning. PMLR, 2017.

**Questions:**

• What are other factors that influence task generalization and why is “reachable task” the most important among these?

• Can you elaborate on a motivating example for this work?

• How do you plan to identify reachable states?

• In comparisons, if you have learned suboptimal tasks for baselines then how is it a fair comparison?

In Figure 3, can you explain the following:

-Why does TEE take longer to train

-If running for more frames, can you show that the performance of TEE matches explore-Go? Are the hyper-parameters chosen such that TEE performs the best it can and are those consistent across all comparisons?

---

### Official Review · Reviewer_SpGy · 2024-11-03

**Soundness:** 3
**Presentation:** 3
**Contribution:** 2
**Rating:** 5
**Confidence:** 4

**Summary:**

The paper tackles the challenge of generalization in multi-task RL by introducing reachability as a key concept. It proposes Explore-Go, a method that enhances generalization by prioritizing training on reachable tasks—those with shared states across tasks—rather than relying solely on extensive exploration. Explore-Go integrates episodic exploration at the start of each episode to improve state diversity, supporting both on-policy and off-policy RL algorithms. The authors validate Explore-Go across discrete and continuous environments, showing improved generalization over baselines like PPO, DQN, and TEE. Their experiments and ablation studies indicate that this approach effectively enhances coverage and policy performance.

**Strengths:**

**Originality**
The paper’s main contribution is its demonstration that training on reachable tasks, rather than merely extending exploration, is more effective for generalization in multi-task RL. This insight, built on the concept of reachability, shifts the focus from continuous to episodic exploration. Reachability is positioned as a form of implicit data augmentation, enhancing generalization to previously unreachable tasks. This approach brings new understanding to why structured exploration aids generalization, addressing a significant gap in multi-task RL.

**Quality**
Explore-Go is implemented effectively across both on-policy and off-policy algorithms, which is notable given the differing requirements of these approaches. The experimental design covers discrete and continuous environments (Four Rooms, DeepMind Control Suite) and testing both state-based and image-based tasks. The paper includes a thorough ablation study (Figs. 8–11) that evaluates the impact of various exploration strategies on metrics like state-action coverage and buffer diversity, strengthening confidence in the method’s robustness across diverse configurations.

**Clarity**
The paper presents its concepts and empirical findings clearly, progressing logically from conceptual definitions to practical results. It effectively situates itself within the context of related work, explicitly contrasting Explore-Go with approaches like Go-Explore, highlighting how Explore-Go focuses on episodic rather than state-coverage-driven exploration.

**Significance**
This work provides valuable insights into why exploration benefits generalization, tackling a key challenge in RL. By demonstrating that training on reachable tasks can drive generalization to unreachable ones, the paper presents a conceptually straightforward yet broadly applicable approach that integrates well with major RL algorithms. The consistent improvements across multiple algorithms (PPO, DQN, SAC) highlight its practical relevance and suggest strong potential for broader applications in multi-task RL.

**Weaknesses:**

**Theoretical limitations**
While the paper provides some theoretical grounding via Corollaries 0.1 and 0.2 (Appendix B), there is no formal analysis of optimal exploration strategies within the reachability framework. This omission leaves open questions regarding optimality, especially in environments with minimal task overlap, where reachability could be limited. Without a formal definition or bounds on reachability, the framework’s applicability to scenarios with sparse or non-overlapping states between training and test tasks is less clear. This limitation could impact generalization performance when the task diversity in the environment is low.

**Methodological issues**
The reliance on simple random exploration (ϵ-greedy with ϵ = 1) and fixed exploration steps (K) offers practical simplicity but does not leverage more adaptive strategies, such as intrinsic motivation or environment-specific adjustments. While the paper acknowledges these choices, adaptive exploration (e.g., adjusting exploration length based on environment complexity) could enhance performance, particularly in complex environments where uniform random exploration may fail to reach key states. The fixed exploration duration restricts scalability to environments where task dynamics or complexities vary widely across episodes.

**Experimental limitations**
The experiments cover standard benchmarks but are limited in terms of task diversity and complexity. While the paper reports no generalization gap in Cheetah Run and Walker Walk (Fig. 12), this reflects properties of these environments rather than a shortcoming of Explore-Go. Nevertheless, testing on more structurally diverse tasks—such as those with distinct reward structures or state transitions—would strengthen the validation of the reachability framework. The lack of comparisons with sophisticated exploration baselines, particularly those using intrinsic rewards, also limits a comprehensive view of the approach’s competitive edge.

**Incomplete validation**
The paper demonstrates strong results in tasks with structural overlap across states, but it remains untested in heterogeneous tasks with distinct goals or dynamics. Real-world applications often involve complex environments with visual and functional variability, where reachability could be inherently low. Testing in such settings would better substantiate the framework's claim to generalization across unseen tasks. Additionally, validation on multi-step dependencies or tasks with sub-goals, which are absent here, could further reveal limitations in the reachability framework’s adaptability to real-world scenarios.

**Questions:**

Theoretical limitations

1. Reachability is defined as the set of states accessible from training tasks, but specific metrics or bounds to quantify reachability in cases of limited overlap are not provided. Could you propose metrics or theoretical bounds for reachability, particularly in settings where training and test tasks share limited overlap? This would clarify reachability’s theoretical applicability across diverse structures.

2. While the reachability framework assumes some degree of task overlap for generalization, the required level of overlap isn’t quantified. Could you specify empirical thresholds or criteria for task overlap that would be necessary to maintain effective reachability, and share any initial findings you might have on this aspect?

Methodological issues

3. The paper uses fixed exploration lengths ($K = 60$ for Four Rooms and $K = 200$ for DeepMind Control Suite), chosen based on visual inspection of coverage, but there is no analysis of Explore-Go’s sensitivity to $K$ or the potential benefits of adapting $K$ dynamically. Could you provide insight into whether varying $K$ might affect performance or generalization, and share any initial observations on how an adaptive $K$ might work based on task complexity?

4. Explore-Go is compared to standard baselines like DQN, PPO, and TEE, but not to more advanced exploration strategies such as RIDE or NovelD, which were cited in the related work. Could you provide your opinion on how Explore-Go might perform relative to these baselines, and what particular strengths or limitations Explore-Go might have compared to intrinsic motivation approaches?

5. The paper notes that adding pure exploration experience to the replay buffer degrades performance (Fig 8), but does not explain why. Could you provide insights into why this degradation occurs, and suggest whether certain forms of this exploration data might be leveraged effectively without negatively impacting performance?

6. Explore-Go relies on initial episode exploration to identify diverse starting states, which may be challenging in environments with sparse rewards or deceptive local optima. Could you discuss how Explore-Go might be adapted for these challenging conditions, where exploration may not naturally find critical states?

Experimental limitations and further validation

7. The paper’s experiments focus on environments where training and test tasks share structural similarities, which may limit insights into the reachability framework’s robustness in low-overlap settings. Could you conduct additional experiments in environments with minimal structural overlap, such as tasks with unique goals, rewards, or distinct starting conditions? This would help substantiate the generalization claims of Explore-Go in scenarios where shared task structure is minimal.

8. The results indicate that training on more reachable tasks enhances generalization to unreachable tasks, likely due to implicit data augmentation. Could you analyze which specific properties of the reachable tasks (e.g., similarity in sub-goals or distribution of states) are most important for enabling this transfer effect?

9. The results show that TEE explores a broader range of states but generalizes less effectively than Explore-Go (Fig 4). Could you discuss why broader state coverage does not necessarily lead to better generalization in this context, and provide any additional analysis you have on why Explore-Go’s method offers advantages over TEE?

10. Although DeepMind Control Suite environments involve some multi-step dependencies, they are relatively simple compared to hierarchical tasks with layered sub-goals. Could you share your perspective on whether Explore-Go’s reachability framework might extend to more complex hierarchical tasks (e.g., those with long-horizon dependencies), and whether adaptations might be needed to make this possible?

11. The paper briefly mentions Procgen environments as a benchmark but does not fully explore specific aspects of real-world variability, such as dynamic visual changes, lighting, or obstacles. Could you provide insight into Explore-Go’s robustness under these conditions and discuss how it might perform in environments with significant visual and structural heterogeneity?

12. For the DeepMind Control Suite experiments, you present results for both state-based and image-based observations. Could you share your perspective on how the reachability concept might differ in these two settings, and whether distinct exploration strategies might be beneficial for each?

---

### Official Review · Reviewer_GpPM · 2024-11-04

**Soundness:** 3
**Presentation:** 3
**Contribution:** 2
**Rating:** 6
**Confidence:** 4

**Summary:**

This paper introduces Explore-Go, a novel method aimed at improving generalization in zero-shot policy transfer (ZSPT) for RL agents by enhancing exploration to increase the number of reachable tasks during training. Precisely, Explore-Go extends any traditional RL environment by including at initial pure exploration (PE) phase at the start of each episode. The samples collected during this PE phase are not used for training. PE is mainly to increase the probability of training from new start states.  By introducing the PE phase at the start of each episode, the authors aim to create agents that generalize better to both reachable and unreachable tasks (defined solely by reachable vs unreachable states). Through various experiments in benchmark environments, including Minigrid and the DeepMind Control Suite, the authors demonstrate that Explore-Go improves testing performance across various RL algorithms without affecting training performance.

**Strengths:**

- **Originality**:
  - The paper proposes a creative solution to generalization in RL by leveraging exploration to maximize reachability of tasks. The exploration phase before each episode adds a distinct approach to state exploration, directly improving sample-efficiency (test performance on reachable states) and zero-shot in-distribution generalization (test performance on unreachable states).
  - While the approach simple and it is generally known in optimization that randomizing the starting state helps with both sample efficiency and generalization because of the exploration benefit, to the best of my knowledge ExploreGo is novel in the way it incorporates that idea into the current RL pipeline.

- **Quality**: The authors provide extensive empirical evaluation of their approach, comparing Explore-Go with baseline methods across several benchmarks and RL algorithms.  Figure 4 (in addition to Figure 3) really nicely demonstrate the effect of different exploration approaches on the diversity of the Replay buffer and sample efficiency.

- **Clarity**: The paper clearly outlines the motivation behind reachability and its implications on generalization, with supportive figures and examples. Figure 1 was especially helpful here. The implementation of Explore-Go is well-detailed (with pseudo-code included in the Appendix), making the method accessible for replication and application.

- **Significance**: The study addresses a pressing issue in RL—sample-efficiency and generalization to unseen tasks—making significant contributions toward more versatile agents. The concept of increasing the diversity of reachable states via pure exploration adds a valuable dimension to current RL methods.

**Weaknesses:**

- **Framework**:
  - The authors have stated throughout the paper and even in the title that they are interested in improving generalization in RL (ZSPT). However, I don't really see anything about the proposed approach that specifically addresses generalization (other than the use of function approximation in the experiments). The proposed approach speaks more to sample-efficiency, and I think the paper would have benefited from making that the focus. For example, I expect similar results (on the training and reachable set) even in the tabular case (e.g with Q-learning).
  - For better clarity, the abstract (and maybe the title too) should have mentioned that the paper is focused on CMDPs, since RL tasks can be more general and usually refer to different reward functions (and/or dynamics) instead of just different contexts.

- **Experiment Limitations**:
  - While the paper is interested in generalization, they include states that were seen during training when evaluating on the reachable test set. This only evaluates sample-efficiency and not generalization.
  - The paper could benefit from experiments comparing Explore-Go to recent state-of-the-art methods designed specifically for generalization and out-of-distribution performance in CMDPs. For example [1] (their domains are also more relevant to CMDPs).
  - While reachability is well-defined in the experiments, its application in real-world, high-dimensional spaces may be limited. The proposed method could face challenges in more complex environments where task states are challenging to categorize as reachable or unreachable. Especially when random exploration can easily lead to a sub-space of states with no returns (e.g. the Minigrid tasks in [2] Fig. 1).
  - In a significant number of experiments, the test set performance of ExploreGo seem to start declining in the second half of training while the baseline keeps increasing. E,g, DQN+TEE 0.1 in Fig 3. It is a problem if this trend actually continues such that the baseline converges to a higher value than ExploreGo.
  - Additionally, DQN+TEE 0.1 in Fig 4.d seem to be doing exactly what it should do and is having the best performance, since it is converging to the optimal policy the fastest.

- **Exploration Strategy**:
  - The uniform exploration phase may not be optimal in all cases. For example it could easy lead agents to unsafe states even if the current policy has already learned the safe behavior.
  - A curiosity-driven or state-novelty-based exploration strategy could enhance sample efficiency and generalization further. While the authors mentioned this as a possibility and provided pseudo-code, this was never evaluated.

  - The exploration parameter K (maximum steps in the PE phase) could be critical to the method's performance, yet there is little discussion on the effect of this parameter across environments. This seems like the biggest limitation of the proposed approach, as tunning this hyperparameter may be as hard as the original exploration-exploitation problem.

[1] Beukman, Michael, et al. "Dynamics generalisation in reinforcement learning via adaptive context-aware policies." Advances in Neural Information Processing Systems 36 (2024).

[2] Vaezipoor, Pashootan, et al. "Ltl2action: Generalizing ltl instructions for multi-task rl." International Conference on Machine Learning. PMLR, 2021.

**Questions:**

Please refer to the weaknesses above. In addition,
- How would "Explore-Go" behave in a four rooms domain with lava states that terminate the episode? E.g.
- Can you run the experiment in Fig 3 for at least 1 million frames to see if the trend continues? And also Fig 8 if possible.
- Can you plot the policy optimality of PPO and PPO-ExploreGo in the four-rooms domain (i.e. similar to Figure 4.d)?
- How sensitive is "Explore-Go" to the choice of maximum exploration steps (K)? Does this parameter need tuning for different environments?
- How are the returns per frames obtained for the four-rooms figures? For example for Fig 3, is it the evaluation return of a single model averaged over 50 evaluation episodes? Or is it the evaluation return of 50 models over a single evaluation episode? Or something in between? Ideally the random seeds should be both for the full training process and evaluation episodes (i.e averaged over multiple training runs and evaluation episodes in each run)
- Do the test-set evaluations of ExplorGo also include PE at the beginning of each evaluation episode?
- There are two DQN+Explore-Go in the caption of Fig 8

---

### Author Response · Authors · 2024-11-26

Unfortunately, during the rebuttal, we discovered an issue with some of our experiments that could not be fixed within the allotted time. Therefore, we have decided to withdraw the paper, fix the problems, and resubmit at a later date. We want to thank all the reviewers for their insightful feedback and suggestions, which we will use to improve the paper.

---

### Note · Authors · 2024-11-26

I have read and agree with the venue's withdrawal policy on behalf of myself and my co-authors.